# Identification of Bioactive Compounds from Marine Natural Products and Exploration of Structure-Activity Relationships (SAR)

**DOI:** 10.3390/antibiotics10030337

**Published:** 2021-03-22

**Authors:** Mojdeh Dinarvand, Malcolm Spain

**Affiliations:** 1School of Chemistry, The University of Sydney, Sydney, NSW 2006, Australia; malcspain@hotmail.com; 2Department of Infectious Diseases and Immunology, Faculty of Medicine and Health, The University of Sydney, Sydney, NSW 2006, Australia; 3Systems Biology, Faculty of Science, School of Biotechnology and Biomolecular Sciences, University of New South Wales, Sydney, NSW 2052, Australia

**Keywords:** marine natural products, tandem mass spectrometry (MS), structure—activity relationships (SAR), antimicrobial drug discovery, multidrug resistant bacteria, methicillin-resistant *Staphylococcus aureus*

## Abstract

Marine natural products (MNPs) have been an important and rich source for antimicrobial drug discovery and an effective alternative to control drug resistant infections. Herein, we report bioassay guided fractionation of marine extracts from sponges *Lendenfeldia*, *Ircinia* and *Dysidea* that led us to identify novel compounds with antimicrobial properties. Tertiary amines or quaternary amine salts: aniline **1**, benzylamine **2**, tertiary amine **3** and **4**, and quaternary amine salt **5**, along with three known compounds (**6–8**) were isolated from a crude extract and MeOH eluent marine extracts. The antibiotic activities of the compounds, and their isolation as natural products have not been reported before. Using tandem mass spectrometry (MS) analysis, potential structures of the bioactive fractions were assigned, leading to the hit validation of potential compounds through synthesis, and commercially available compounds. This method is a novel strategy to overcome insufficient quantities of pure material (NPs) for drug discovery and development which is a big challenge for pharmaceutical companies. The antibacterial screening of the marine extracts has shown several of the compounds exhibited potent in-vitro antibacterial activity, especially against methicillin-resistant *Staphylococcus aureus* (MRSA) with minimum inhibitory concentration (MIC) values between 15.6 to 62.5 microg mL^−1^. Herein, we also report structure activity relationships of a diverse range of commercial structurally similar compounds. The structure-activity relationships (SAR) results demonstrate that modification of the amines through linear chain length, and inclusion of aromatic rings, modifies the observed antimicrobial activity. Several commercially available compounds, which are structurally related to the discovered molecules, showed broad-spectrum antimicrobial activity against different test pathogens with a MIC range of 50 to 0.01 µM. The results of cross-referencing antimicrobial activity and cytotoxicity establish that these compounds are promising potential molecules, with a favourable therapeutic index for antimicrobial drug development. Additionally, the SAR studies show that simplified analogues of the isolated compounds have increased bioactivity.

## 1. Introduction

Human pathogens are associated with a variety of moderate to severe infections and the recent rise of multi-drug resistant pathogens makes treatment more difficult [1]. The last two decades have seen the emergence of methicillin-resistant *Staphylococcus aureus* (MRSA) strains resistant even to ‘drugs of last resort’ such as vancomycin [2], and *Mycobacterium tuberculosis* resistant to all first-line agents [3,4,5], highlighting the urgent need to find new effective antibiotics with distinct mechanisms of action.

Natural products continue to offer a productive source of structural diversity and bioactivity and therefore hold the potential for the discovery of new and efficacious antimicrobial drugs [6,7,8,9]. Almost 70 percent of the Earth’s surface is covered by ocean, representing a huge reserve of natural biological and chemical diversity on our planet [10]. Marine ecosystems have long been a rich source of bioactive natural products in the search for interesting molecules and novel therapeutic agents [6,7,11,12,13]. Many interesting and structurally diverse secondary metabolites have been isolated from marine sources over the last 70 years [8,9,14,15]. In addition, the preclinical pharmacology of seventy-five compounds isolated from marine organisms have been reported to have biological activities [16]. Yet the first drugs from marine natural products were only approved in the early 2000s: the cone snail peptide ziconotide (ω-conotoxin MVIIA) in 2004 to alleviate chronic pain [17], and sea squirt metabolite trabectedin in 2007 for the treatment of soft-tissue sarcoma [18]. Marine natural products (MNPs) have displayed exceptional potency and potential as anticancer therapeutics [19]. The interest in MNPs has continued to grow [8,14,15], spurred in part by the spread of antimicrobial resistant pathogens and the need for new drugs to combat them [6].

The most prolific marine organisms are sponges [6], and the oldest metazoans on earth belong to the phylum *Porifera* [20]. The Demospongiae are the most abundant class of Porifera, representing 83% of described species [20,21], and has the largest number of bioactive compounds [19]. The genus *Lendenfeldia* is known as a source of sulfated sterols [22] and metabolites from the *Lendenfeldia* species have anti-HIV, anti-tumor [23], anti-inflammatory, antifouling [24] activities but they lack antimicrobial activity [22]. Secondary metabolites of the genus *Ircinia* and *Dysidea* are prime candidates for further study to unveil biological metabolites with antibacterial activity (Figure 1) [15,25,26,27].

In the search for new antimicrobial agents, we screened a set of marine extracts [28] to determine activity against antibiotic resistant microorganisms using a high-throughput screening (HTS) assay. The fractionation and purification of active components by high-performance liquid chromatography (HPLC), Nuclear magnetic resonance (NMR) and structural elucidation using high resolution and tandem mass spectrometry (MS) led us to a series of potential scaffolds for new, bioactive amine natural products (Figure 1 and Appendix A).

Following on from these studies, we decided to screen commercially available compounds, with structural similarity to the amine scaffolds we identified, synthesised [29] and ascididemin-like compounds. Thus, we found these molecules to be an interesting potential structure for further development, focusing on increasing antimicrobial potency by investigating the structure—activity relationship (SAR). Based on these facts, in this work, small-molecule libraries of commercial analogues were studied to increase the structural variability and improving the pharmacological properties of the final drugs.

## 2. Results and Discussion

### 2.1. Identification of Active Marine Extracts

To identify marine samples with activity against MRSA, 1434 compounds from the AIMS Bioresources Library [28] (provided by the Queensland Compound Library, Brisbane, Australia [30], now called Compounds Australia [31]) were screened in a resazurin cell viability assay. Of the samples tested, 29 inhibited the growth of MRSA by greater than 50% compared to non-treated controls. The minimum inhibitory concentrations (MICs) were determined for the 23 most promising samples, representing extracts and fractions from the phyla Porifera (90%), Echinodermata (5%) and Chordata (5%) (Table 1). The five most active samples showed MICs at 31.3 µg mL^−1^ (all Porifera samples), while another four samples returned MICs of 62.5 µg mL^−1^ (also all Porifera). Cytotoxicity screens against HepG2, HEK 293, A549 and THP-1 cell lines were performed with concentrations from 500 µg mL^−1^ to 7.8 µg mL^−1^ to define the cytotoxicity profile of the most active samples. Pleasingly, all the samples most active against MRSA were also nontoxic to the cell lines tested (Table 1).

### 2.2. Isolation and Characterization of Bioactive Compounds

Following the primary screening of the AIMS library and selection of positive hits, HPLC was used to separate and isolate active compounds, guided by bioassays against MRSA. The extracts were fractionated by analytical HPLC (see Experimental section and Appendix A for further details), and fractions evaluated for bioactivity. Preparative scale HPLC was carried out on each bulk sample to isolate the active component (Appendix A), NMR and tandem mass spectrometry (MS/MS) methods used to deduce potential structures for compounds (Table 2 and Appendix A) [32,33,34]. Insufficient quantities were obtained for positive ion high resolution mass spectrometry (HRMS) analyses.

Active components were isolated and characterised for eight of the nine extracts shown in Table 1: aniline **1** and benzylamine **2** from the *Lendenfeldia* sp. samples (AIMS Sample Code 20608, Table 1 entries 2–4, Figure 2); tertiary aliphatic amine **3** from the *Dysidea herbacea* extract (AIMS Sample Code 19033, Table 1 entry 1, Figure 2); aliphatic tertiary amine **4** and quaternary amine salt **5** from *Ircinia gigantea* (AIMS Sample Code 26051, Table 1 entry 5, Figure 2); ascididemin **6** and 2-bromoascididemin (2-Bromoleptoclinidinone)**7** from the *Flavobranchia* samples (AIMS Sample Code 25663, Table 1 entries 7, Figure 2 and Appendix A); and halisulfate **8** from Ircinia (AIMS Sample Code 26104, Table 1 entry 8, Figure 2 and Appendix A). The aniline/amines **1**–**5** have not previously been identified as natural products, nor have their antimicrobial activities been assessed.

Compounds **6**, **7** [35,36,37] and **8** [38,39,40,41] have previously been reported (Figure 2), and in the current study these structures were confirmed by comparison of data with HRMS, NMR and literature values. The antimycobacterial activity of compounds **6, 7** and **8** with wide range of MIC against different microorganisms have also been reported [42,43].

### 2.3. Structure—Activity Relationship

The SAR was investigated through commercially available compounds of three groups of molecules, namely the tertiary amines, benzyl substituted amines [29], and the ascididemin structural motifs. To investigate the number of carbons in the linear chains, commercially available amines compounds with structures similar to synthesised compounds **3** and **13** [29] (Figure 3), such as triethylamine, tributylamine and trihexylamine were screened; in addition, the primary and secondary amines hexylamine, dihexylamine and dioctylamine were screened. Biological activities of commercial compounds are based on the molecular weight of the proposed structures.

The results show a general trend of increased activity with increasing chain length of the tertiary amines (triethylamine, tributylamine, trihexylamine and trioctylamine), as seen in the bioactivity against MRSA, *E. coli* and *P. aeruginosa* (Table 3). In addition, the trend in the primary, secondary, and tertiary hexylamines, show increased activity with an increasing number of alkyl chains; hexylamine, showed activity against MRSA, *E. coli* and *P. aeruginosa* with MICs of 50 to 100 µM, which all improve to MICs of 3 to 6.5 µM for trihexylamine (Table 3). Interestingly, trioctylamine with eight carbon alkyl chains, showed strong activity against *M. tuberculosis* H37Rv (MIC of 0.02 µM). Furthermore, dicyclohexylamine has poorer bioactivity than dihexylamine, suggesting potential benefits of linear chains. This trend in structural modification did not improve activity against *M. tuberculosis* H37Rv or alter the toxicity to cell lines except HEK 293 (Table 3).

Further structures related to 1a, 1b and 2a [29] and the sequence of benzylamine, dibenzylamine, tribenzylamine, were investigated (Figure 4). Observations from this experiment correlate with the increased substitution leading to increased bioactivity against *P. aeruginosa*. Anti *P. aeruginosa* activity was improved to MIC = 6.5 with tribenzylamine (Table 4).

The third group of compound structures we looked at for SAR analysis are related to the structural motifs of ascididemin (Figure 5), which is a pentacyclic aromatic alkaloid, bearing similarity to motifs of quinoline and phenanthroline. Therefore, isoquinoline, quinoline, 4,7-phenanthroline, 1,7-phenanthroline, 1,10-phenanthroline and bipyridine structures (Table 4 and Table 5) were evaluated for bioactivity inspired by the ascididemin structure. The results show isoquinoline and quinoline displayed improved anti-microbial activity, especially against *M. tuberculosis* (Table 5). Furthermore, the fused tricyclic cores of 1,10-phenanthroline, when compared with the more flexible 2,2′-bipyridine, showed activity against *M. tuberculosis* with MIC = 0.1 but also displayed toxicity for 4 cell lines tested: HepG2, HEK 293 and THP-1. Interestingly, structures in which the two nitrogens were closer to each other have more activity against *M. tuberculosis* and *P. aeruginosa* while the opposite situation was seen with MRSA and *E. coli* activity; this can be observed by comparing the results of 1,10-phenanthroline with those observed for 4,7-phenanthroline. The toxicity assays determined that the A549 cell line is more resistant to 4,7-phenanthroline. Alongside the acceptable antimicrobial activity of 4,7-phenanthroline and 1,7-phenanthroline, also show limited toxicity against mammalian cells (Table 5). Meanwhile, 1,10-phenanthroline displayed variable toxicity and antimicrobial activity (Table 5).

To further study the phenanthroline compounds (Figure 6) we evaluated 4,7-dimethyl-1,10-phenanthrolin, neocuproine hemihydrate, 5,6-dimethyl-1,10-phenanthroline, 3,4,7,8-tetramethyl-1,10-phenanthroline, 1,10-phenanthroline-5,6-dione, 5-nitro-1,10-phenanthroline, bathophenanthroline, 2,9-dimethyl-4,7-diphenyl-1,10-phenanthrolin, and bathocuproine disulfonic acid for bioactivity (Table 6). Modifying the three aromatic rings further, by adding methyl groups as in 4,7-dimethyl-1,10-phenanthroline, neocuproine hemihydrate and 5,6-dimethyl-1,10-phenanthroline (Figure 7) improved the anti-bacterial activity of the compounds but also increased their toxicity effect on all cell lines tested (Table 6). Maintaining one of the methyl groups in the rings as in 5,6-dimethyl-1,10-phenanthroline reduced the activity against *P. aeruginosa* (Table 6).

Increasing the number of methyl substituents to four (3,4,7,8-tetramethyl-1,10-phenanthroline) or incorporating carbonyl groups (1,10-phenanthroline-5,6-dione) or a nitro group (5-nitro-1,10-phenanthroline) instead of the methyl groups enhanced the activity of the compounds (Table 6 and Table 7). A different behaviour was only observed against *P. aeruginosa* in the presence of the nitro group. The inclusion of two additional aromatic rings as pendants on the phenanthroline core (bathophenanthroline) showed impressive antibacterial activity with MIC = 0.09 µM for MRSA, MIC = 0.019 µM for *E. coli*, MIC = 0.03 µM for *P. aeruginosa* and MIC = 0.01 µM for *M. tuberculosis*. Furthermore, cross-referencing the biological activity and toxicity of these molecules demonstrated they are more potent against the microorganisms than against mammalian cells. In addition, adding two methyl groups or two sulfonate groups, as in 2,9-dimethyl-4,7-diphenyl-1,10-phenanthroline and bathocuproine disulfonic acid, respectively, resulted in the reduced activity of molecules against bacteria, but also a reduction in toxicity against the A549 and THP-1 cells. However, including the sulfonate groups improved the activity against the *P. aeruginosa* with MIC = 0.19 µM (Table 7).

A summary of all activities of the commercially available amines that were screened is presented in Table 8. The potential antimicrobial activity of commercial materials which are structurally related to ascididemin (Table 6 and Table 7) was identified by bioactivity screening. 1,10-phenanthroline-5,6-dione at MIC 0.04 μM was found to be a potent inhibitor of MRSA, while bathophenanthroline at MIC 0.019 μM, MIC 0.03 μM and MIC 0.01 μM showed strong activity against *E. coi*, *P. aeruginosa* and *M. tuberculosis,* respectively. These results reinforce the relationship between structures and anti-microbial activity for the synthesised compounds. For example, in the third group of compounds tested (Table 6 and Table 7) bioactivity was optimal for 1,10-phenanthroline-5,6-dione containing carbonyl groups, and biological activity depends on structural properties such as the length of the aliphatic linker, and the side chain of these compounds. The results of these experiments further suggest that hydrophobic compounds, such as dioctylamine (Table 3) and bathophenanthroline (Table 7), have a more pronounced effect on the inhibition of bacterial growth, as has been reported previously [41,42,43,44,45,46,47].

Furthermore, the side chain length significantly affects the hydrophobicity of these compounds, and subsequently the logP values, which are important molecular descriptors in determining biological activity. The presence of aromatic rings with short pendant groups and nitrogen-containing rings in 2,2′-bipyridine significantly reduced the activity of the molecules against the Gram-positive and Gram-negative pathogens tested, however the compounds were active against mycobacteria (Table 4). This suggests that the mechanism of action may be based on targeting complex cell wall structures, such as those present in mycobacteria. Many factors influence the activity of a molecule, including side chain length, symmetry, length of the alkyl chain, the presence of charged pyridinium moiety, the presence and position of the carboxylate in the structure, the position of the methyl group, the distance between the two pyridinium moieties and the chain linker between the two pyridinium rings. Thus, the mechanism of action of these molecules may be based on the space between the rings and distance between charged moieties, which possibly manifests through interaction with membrane sites. These results are in agreement with previous reports [48,49]; however, to clarify such hypotheses and determine important factors in bioactivity further pharmacological and biophysical studies are necessary to determine the mechanisms of action.

## 3. Materials and Methods

### 3.1. General

Chemical reagents were purchased from BDH Chemicals and Sigma Aldrich (Castle Hill, Sydney, Australia) and used as supplied unless otherwise indicated. Commercial amines of >97% purity were purchased from Sigma Aldrich and were used in this study (CAS numbers are added in the Appendix A).

### 3.2. Natural Product Library

Natural product extracts were provided by the Australian Institute of Marine Science (AIMS), Townsville, Australia as part of the AIMS Bioresources Library [28], via the Queensland Compound Library, Brisbane, Australia [29], (now called Compounds Australia [30]). Crude extracts had been partially fractionated by AIMS/ QCL to generate a library of 1434 samples, supplied in DMSO (100%) solution and stored at −80 °C. Original concentrations as provided were 5 mg mL^−1^. Stock solutions were made by diluting these samples by a factor of 1:10 in dH_2_O and stored at −80 °C.

### 3.3. Bacterial Inhibition Assays

For screening of the AIMS library, each test sample (10 µL) was dispensed into a separate well of a 96 well microtiter plates (final sample concentration 0.5 mg mL^−1^) using sterile dH_2_O. For determination of MIC, extracts (250 to 0.5 µg mL^−1^) or synthesised compounds (100 to 0.0002 µM) were serially diluted in microtiter plates. Bacterial suspension (90 µL, OD_600nm_ 0.001) was added to each well and plates were incubated at 37 °C for either 18 h (MRSA, *P. aeruginosa* PAO1, *E. coli* EC958) or 7 days for *M. tuberculosis* H37Rv as described previously [50]. Resazurin (10 μL; 0.05% *w*/*v*) was added and plates were incubated for 3 h or 24 h (*M. tuberculosis*) at 37 °C. The inhibitory activity was calculated by visual determination of colour change within wells or detection of fluorescence at 590 nm using a FLUOstar Omega microplate reader (BMG Labtech, Ortenberg, Germany). Percentage survival was calculated in comparison to the average of untreated control wells after normalising for background readings.

### 3.4. Evaluating

Toxicity of AIMS Extract Library. Human alveolar epithelial cells (A549) [51], Madin-Darby canine kidney epithelial cells (MDCK) [52], human leukaemia cells (THP-1) [53], human hepatocellular carcinoma cells (Hep-G2) [54] and human embryonic kidney cells 293 (HEK293) [55] were grown and differentiated in complete RPMI (Roswell Park Memorial Institute Medium) and DMEM (Dulbecco’s Modified Eagle’s medium) tissue culture media (RPMIc and DMEMc, from the Thermo Fisher Scientific, Waltham, MA, USA) by adding 10% FBS, 200 µM L-glutamine and 1 mM HEPES buffer solution. To determine toxicity of the AIMS extract library, 2 × 10^5^ of each cell type were added to a 96-well plate and left for 48 h in a humidified 5% CO_2_ incubator at 37 °C to adhere. Extract samples at a final concentration of 0.5 mg mL^−1^ were added to the wells, then the plates incubated for 4 more days in a humidified 5% CO_2_ incubator at 37 °C. Then, resazurin (10 μL of 0.05% *w*/*v*) was added and after 4 h, fluorescence measured as described previously. The cell viability was calculated as percentage fluorescence relative to untreated cells.

### 3.5. Purification of Natural Products from Extracts and Structure Elucidation

#### 3.5.1. High Performance Liquid Chromatography (HPLC) Purification

The samples were separated using analytical (Waters 2695 Alliance pump with Waters 2996 PDA, Sunfire reversed-phase column, and WFIII fraction collector) and preparative (Waters 600 HPLC pump, Phenomenex reversed-phase column, Waters 2487 UV detector and WFIII fraction collector) HPLC systems with UV detectors at 254 and 280 nm, employing a gradient of solvents A (dH_2_O) and B (acetonitrile) with trifluoroacetic acid (0.1%). Extract mixtures were kept at 4 °C until injection, then extract sample (100 μL) was injected onto an analytical Waters X-bridge C18 100 Å (4.6 × 250 mm, 5 µm) reversed-phase column on the same analytical HPLC system described above. The mobile phase was obtained using binary gradients of solvents A and B at a flow rate of 1 mL min^−1^ at 30 °C over 80 min. Fractions, separated every 60 s, were collected. The purified fractions were flash-frozen in liquid nitrogen then freeze-dried overnight. The resulting fractionated extracts were re-suspended in DMSO and antibacterial activity versus MRSA was determined as described above. Fractions identified as active against MRSA were further purified on the preparative HPLC unit described above, using a Phenomenex C18 100 Å (250 × 21.2 mm, 10 µm) reversed-phase column with UV detection at 254 and 280 nm, 7 mL min-1flow rate with water/acetonitrile gradient containing 0.1% trifluoroacetic acid. The gradient for AIMS extracts 19033, 20608, 26104, 25641 and 26051 was 0% B initially, increased to 40% B over 20 min, then to 100% over 40 min, held at 100% for 10 min. For AIMS extracts 25663 gradient was 0% B initially, increased to 40% B over 60 min, then to 100% over 30 min, held at 100% for 10 min. Compounds thus purified were evaluated for biological activity and analysed by MS to determine potential structures for the bioactive components.

#### 3.5.2. Identification and Structure Elucidation

##### Identification and Elucidation of Purified Compounds from Sample Numbers 19033, 20608 and 26051

Purified compounds were identified and characterised using MS and NMR. High resolution ESI mass spectra (HRMS) were recorded on a Bruker Apex Qe 7T Fourier Transform ion cyclotron resonance mass spectrometer with an Apollo II ESI MTP ion source with samples (in CH_3_CN:H_2_O 1:1) infused using a Cole Palmer syringe pump at 180 µL h^−1^. Where required, low resolution ESI tandem MS was performed on a Bruker amaZon SL ion trap via syringe infusion or by injection into a constant flow stream with a rheodyne valve and an Alltech HPLC pump (mobile phase methanol, flow rate 0.3 mL min^−1^) connected to an Apollo II ESI MTP ion source in positive ion mode. Tandem mass spectra of the [M + H]^+^ parent ion were obtained manually up to MS5 (depending on sensitivity). Spectra were acquired in positive ion mode using a 1–4 Da isolation window, with the excitation amplitude manually optimized for each spectrum to have the selected mass at ~10% of the height of the largest fragment. Data analysis was performed for both high resolution MS and low-resolution tandem MS data using Bruker Data Analysis 4.0 with smart formula assuming C, H, N, O, Na (0-1), mass error <2 ppm, C:H ratio 3 maximum, even electron (or both for tandem MS data). The results of high-resolution MS data analysis were further refined manually by comparing isotopic fine structures of simulations where possible (resolving power >200,000) to further eliminate potential formulae within the 2ppm mass error window (particularly ^15^N, ^18^O, ^2^H, ^13^C and ^13^C2 isotopes and confirm no ^34^S presence).

##### Identification and Elucidation of Fractions 44 from Sample Number 25663 (Compound **6**)

^1^H NMR (500 MHz, DMSO-*d*_6_): δ 9.20 (d, *J* = 5.6, 1H, H-21), 9.10 (dd, *J* = 4.5, 1.7, 1H, H-17), 9.00 (dd, *J* = 8.0, 1.1, 1H, H-6), 8.92 (d, *J* = 5.6, 1H, H-19), 8.63 (dd, *J* = 7.9, 1.7, 1H, H-15), 8.44 (d, *J* = 8.0, 1H, H-3), 8.09 (ddd, *J* = 8.0, 7.2, 1.1, 1H, H-2), 8.02 (ddd, *J* = 8.0, 7.2, 1.1, 1H, H-1), 7.80 (dd, *J* = 7.9, 4.5, 1H, H-16). HRMS (ESI): *m*/*z* 306.0635; the molecular formula C_18_H_9_ON_3_ gives an expected molecular [M + Na]^+^ ion at 306.0637 (err 0.8 ppm). This molecular formula gives a rdbe of 16, requiring many rings or double bonds. The NMR and mass data are in agreement with those in the literature for the known compound ascididemin [36,37,55].

##### Identification and Elucidation of Fractions 58 from Sample Number 25663 (Compound **7**)

^1^H NMR (500 MHz, DMSO-*d*_6_): δ 9.25 (d, *J* = 5.3, 1H, H-21), 9.13 (d, *J* = 4.3, 1H, H-17), 8.98 (d, *J* = 8.8, 1H, H-6), 8.93 (m, 1H, H-19), 8.68 (s, 1H, H-15), 8.64 (d, *J* = 7.6, 1H, H-3), 8.19 (d, *J* = 8.8 1H, H-1), 7.81 (dd, *J* = 4.3, 7.6, 1H, H-16). HRMS (ESI): *m*/*z* 361.9925; the molecular formula C_18_H_8_BrN_3_O would give an expected [M + H]^+^ ion at 361.9923 (err 0.5 ppm). This molecular formula gives a rdbe of 16, requiring many rings or double bonds. The NMR and mass data agree with those in the literature for the known compound 2-bromoascididemin [36,37].

##### Identification and Elucidation of Purified Compounds from Sample Number 26104 (Compound **8**)

^1^H NMR (500 MHz, DMSO-*d*_6_): δ 8.47 (br s, 1H, H-8), 8.46 (br s, 1H, H-7), 6.54 (d, *J* = 8.4, 1H, H-6), 6.44 (d, *J* = 3.2, 1H, H-3), 6.35 (dd, *J* =3.2, 8.4, 1H, H-5), 5.29 (m, 1H, H-25), 5.22 (t, *J* = 7.3, 1H, H-10), 4.19 (d, *J* = 11.4, 1H, H-18), 3.12 (d, *J* =7.3, 1H, H-9), 2.19 (m, 1H, H-21), 2.13 (m, 1H, H-16), 1.95 (t, *J* = 7.4, 1H, H-13), 1.91 (m, 1H, H-24), 1.8 (m, 1H, H-24′), 1.66 (d, *J* = 13.1, 1H, H-27), 1.62 (s, 3H, H-12), 1.6 (s, 3H, H-31), 1.42 (m, 1H, H-14), 1.41 (m, 1H, H-28), 1.37 (m, 1H, H-29), 1.34 (m, 1H, H-20), 1.21 (m, 1H, H-15), 1.13 (m, 1H, H-23), 1.12 (m, 1H, H-29′), 1.06 (m, 1H, H-20′), 1.03 (m, 1H, H-27′), 1.03 (m, 1H, H-15′), 0.84 (s, 3H, H-33), 0.83 (s, 3H, H-34), 0.78 (d, *J*= 6.9, 1H, H-26), 0.66 (s, 3H, H-32). LRMS (APCI): *m*/*z* 467.3, [M + H]^+^; HRMS (APCI): *m*/*z* 467.1, [M + H]^+^; the molecular of formula C_30_H_17_N_2_O_4_ calculated at 467.118, err 1.9 ppm. This results in rdbe calculation of 24 indicating a high level of unsaturation and rings. The NMR and mass data agree with those in the literature for the known compound halisulfate [55,56,57,58].

## 4. Conclusions

The emergence of antibiotic resistance highlights the need for novel, effective antibacterial agents which circumvent traditional resistance mechanisms. Thus, we assayed 1434 extracts from the AIMS Bioresources Library, Brisbane, Australia [28] against MRSA, finding five samples that have a promising combination of high antibacterial activity and low toxicity to mammalian cells. The NMR, high-resolution MS and tandem MS analysis was used to decipher structures. The proposed structures form samples 20608, 26051 and 19033 are all tertiary amines or quaternary amine salts: aniline **1** and benzylamine **2** (from *Lendenfeldia* sample number 20608), aliphatic amine **3** (from *Dysidea herbacea* sample number 19033), aliphatic tertiary amine **4** and quaternary amine salt **5** (from *Ircinia* sp. sample number 26,051). The proposed structures from samples 25663 and 26104 are acididemin **6** and 2-bromoascididemin **7** (from *Flavobranchia* sample number 25663), and halisulfate **8** (from *Ircinia* sample number 26104). The compounds discovered in this study add to the growing arsenal of antimicrobial agents from the sea [6,12], and offer interesting new avenues in the quest for new, effective agents to combat the growing scourge of multidrug resistant bacteria.

The structure activity relationship (SAR), close analogues studies have demonstrated active scaffolds of isolated natural products and synthetic derivatives compounds. These natural product analogues may be optimized further, inspiring further studies in the search for new lead compounds to fight against microbial pathogens. They will be helpful in broadening the understanding of the biological effects of the synthetic analogues of ascididemin and therefore may aid the development of novel anticancer and antimicrobial agents. Often the natural properties of natural compounds, such as molecular weight, a large number of chiral centres, and/or complex 3D structures can hamper the drug development process and make them unsuitable for synthesis or non-drug-like [59,60]. Interestingly, the simple molecules which have been found in this study may be suitable as potential drug leads.

## Figures and Tables

**Figure 1 antibiotics-10-00337-f001:**
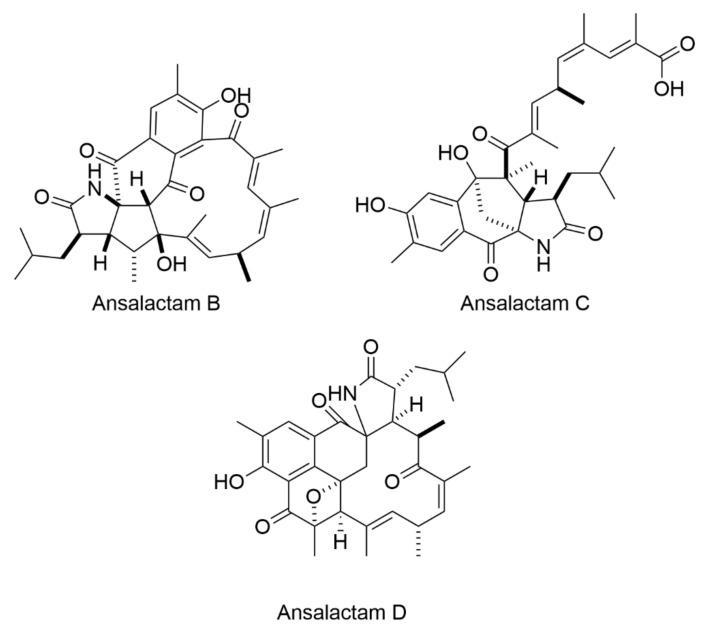
Ansalactams B–D displayed moderate antibacterial activity towards MRSA [15].

**Figure 2 antibiotics-10-00337-f002:**
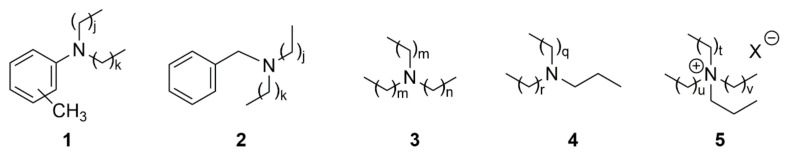
Proposed structures of bioactive amine natural products identified as new natural products compounds (**1**–**5**) and rediscovered compounds (**6**–**8**) in this study; j = 14; m = 5, n = 9; q = 20; t = 19; X = unidentified counterion.

**Figure 3 antibiotics-10-00337-f003:**
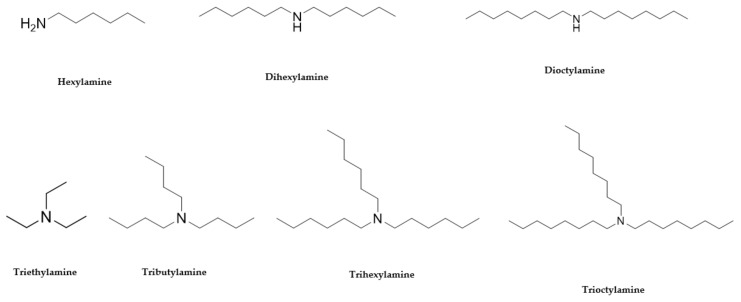
The commercially available amines with structures similar to 3 and 13 [29].

**Figure 4 antibiotics-10-00337-f004:**
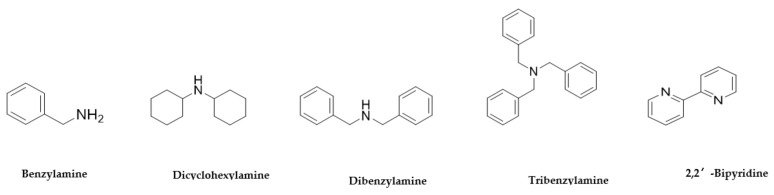
The commercially available amines with structures similar to 1a, 1b and 2a [29].

**Figure 5 antibiotics-10-00337-f005:**
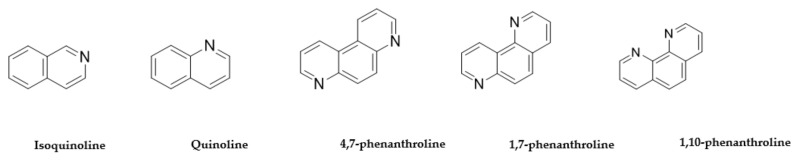
The commercially available compounds with structures inspired by ascididemin.

**Figure 6 antibiotics-10-00337-f006:**
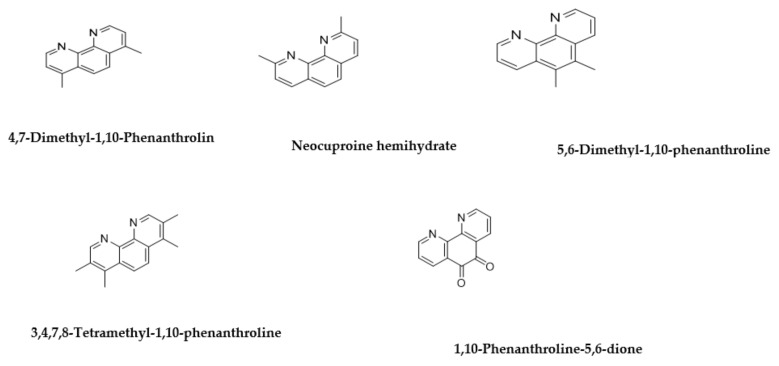
The compounds with structures inspired by ascididemin which include three aromatic rings.

**Figure 7 antibiotics-10-00337-f007:**
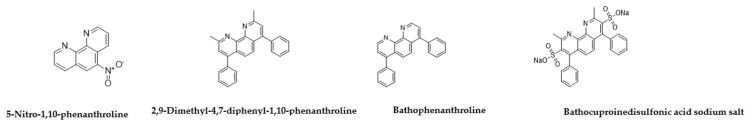
The compounds with structures inspired by ascididemin which include three aromatic rings plus further modifications.

**Table 1 antibiotics-10-00337-t001:** Summary of the nine marine samples selected for further study.

Entry	AIMS Sample Code	QCL Sample Number	MIC (µg mL^−1^)	Cytotoxicity (% Cell Survival)		
			MRSA	Hep G2	A549	HEK
1	19,033	SN00733110	31.25 ± 0.9	91 ± 1.2	91 ± 0.3	98 ± 1.6
2	20,608	SN00760947	31.25 ± 1.3	97 ± 1.0	101 ± 2.9	102 ± 0.7
3	20,608	SN00760956	31.25 ± 0.4	100 ± 4.4	106 ± 1.8	95 ± 1.0
4	20,608	SN00760958	62.5 ± 2.2	98 ± 0.6	108 ± 14	95 ± 2.5
5	26,051	SN00731005	62.5 ± 2.8	101 ± 1.0	110 ± 2.7	98 ± 2.0
6	24,307	SN00730755	31.25 ± 1.0	100 ± 1.0	106 ± 1.0	96 ± 4.0
7	25,663	SN00732222	15.6 ± 1.5	100 ± 1.6	89 ± 2.0	97 ± 0.0
8	26,104	SN00734298	62.5 ± 1.0	98 ± 0.5	100 ± 1.1	98 ± 1.1
9	22,565	SN00739718	31.25 ± 0.4	100 ± 0.0	99 ± 0.4	97 ± 1.0

**Table 2 antibiotics-10-00337-t002:** Key high resolution mass spectrometry (HRMS) data for bioactive samples, and proposed structures as shown in the Appendix A.

Sample	Molecular Ion (m/*z*)	Molecular Formula	Proposed Structures	Spectra	MS/MS	NMR Analyses
19,033 ^†^	326.37813 [M + H]^+^	[C_22_H_48_N]^+^ Calc. = 326.37802 (∆m = 0.11 ppm) RDBE = 0	3	Appendix A	Appendix A Appendix A	-
20,608	332.33115 [M + H]^+^	[C_23_H_42_N]^+^ Calc = 332.33118 (Δm = 0.03 ppm) RDBE = 4 ^‡^	1,2	Appendix A	Appendix A Appendix A	-
26,051	368.42508 [M + H]^+^	[C_25_H_54_N]^+^ Calc. = 368.42495 (∆m = 0.13 ppm) RDBE = 0	4,5	Appendix A	Appendix A Appendix A	-
25,663	306.0635 [M+Na]^+^	[C_18_H_9_ON_3_] Calc. = 306.0637 (∆m = 0.8 ppm) RDBE = 16	6	Appendix A	-	Appendix A Appendix A
25,663	361.9925 [M + H]^+^	[C_18_H_8_BrN_3_O] Calc. = 361.9923 (∆m = 0.5 ppm) RDBE = 16	7	Appendix A	-	Appendix A Appendix A
26,104	467.117 [M + H]^+^	[C_30_H_17_N_2_O_4_] Calc. = 467.118 (∆m = 1.9 ppm) RDBE = 0	8	Appendix A	-	Appendix A Appendix A

† The same active species was observed for all five fractions SN00760947, SN00760956, SN00732222, SN00734298 and SN00760958. ‡ RDBE = ring or double bond equivalents.

**Table 3 antibiotics-10-00337-t003:** Screening of commercially available amines with structures similar to 3 and 13 (The biological activities are based on the molecular weight of the proposed structures).

Name and CAS no.	Structures	MRSA	*E. coli*	*P. aeruginosa*	*M. tuberculosis*	HepG2	HEK 293	A549	THP-1
**Hexylamine**111-26-2	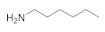	>100	>100	50	>100	>100	>100	>100	>100
**Dihexylamine**143-16-8	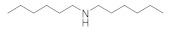	12.5	12.5	6.2	>100	>100	>100	>100	>100
**Dioctylamine**1120-48-5	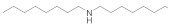	6.2	6.2	3	>100	>100	>100	>100	>100
**Triethylamine**121-44-8	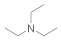	>100	>100	>100	>100	>100	>100	>100	>100
**Tributylamine**102-82-9	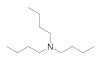	>100	>100	50	>100	>100	>100	>100	>100
**Trihexylamine**102-86-3	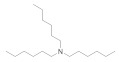	3	3	6.5	>100	>100	50	>100	>100
**Trioctylamine**1116-76-3	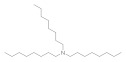	12.5	25	25	0.02	>100	6	50	3.1

**Table 4 antibiotics-10-00337-t004:** Screening of commercially available amines with structures related to 1a, 1b and 2a.

Name and CAS no.	Structures	MRSA	*E. coli*	*P. aeruginosa*	*M. tuberculosis*	HepG2	HEK 293	A549	THP-1
**Benzylamine**100-46-9	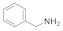	>100	>100	>100	>100	>100	>100	>100	>100
**Dicyclohexylamine**101-83-7	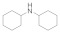	>100	>100	12.5	>100	>100	>100	>100	>100
**Dibenzylamine**103-49-1	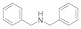	>100	>100	12.5	>100	>100	>100	>100	>100
**Tribenzylamine**620-40-6	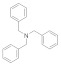	>100	>100	6.5	>100	>100	>100	>100	>100
**2,2′-Bipyridine**366-18-7	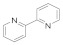	>100	>100	>100	0.1	3	3	>100	50

**Table 5 antibiotics-10-00337-t005:** Screening of commercially available compounds with structures inspired by ascididemin.

Name and CAS no.	Structures	MRSA	*E. coli*	*P. aeruginosa*	*M. tuberculosis*	HepG2	HEK 293	A549	THP-1
Isoquinoline119-65-3	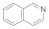	>100	>100	>100	0.1	3	>100	50	3
Quinoline91-22-5	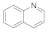	>100	>100	>100	0.1	3	50	50	3
4,7-phenanthroline230-07-9	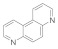	1.5	1.5	2.5	2.5	>100	>100	>100	>100
1,7-phenanthroline230-46-6	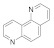	25	25	2.5	>100	>100	>100	>100	>100
1,10-phenanthroline66-71-7	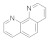	>100	50	0.3	0.1	0.1	50	3	0.1

**Table 6 antibiotics-10-00337-t006:** Screening of compounds with structures inspired by ascididemin which include three aromatic rings.

Name and CAS no.	Structures	MRSA	*E. coli*	*P. aeruginosa*	*TB. H37Rv*	HepG2	HEK 293	A549	THP-1
4,7-Dimethyl-1,10-Phenanthrolin3248-05-3	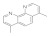	0.3	0.3	0.1	0.1	0.1	0.1	0.1	0.1
Neocuproine hemihydrate34302-69-7	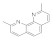	0.09	0.3	0.04	0.1	0.1	0.1	6.2	12
5,6-Dimethyl-1,10-phenanthroline3002-81-1	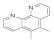	0.3	0.78	>100	0.1	0.1	0.1	0.1	0.1
3,4,7,8-Tetramethyl-1,10-phenanthroline1660-93-1	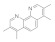	0.3	0.3	0.3	0.1	0.1	0.1	0.1	0.1
1,10-Phenanthroline-5,6-dione27318-90-7	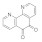	0.04	0.04	0.04	0.1	0.1	0.1	0.1	0.1

**Table 7 antibiotics-10-00337-t007:** Screening of compounds with structures inspired by ascididemin which include three aromatic rings plus further modifications.

Name and CAS no.	Structures	MRSA	*E. coli*	*P. aeruginosa*	*TB. H37Rv*	HepG2	HEK 293	A549	THP-1
5-Nitro-1,10-phenanthroline4199-88-6	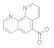	0.09	0.78	>100	0.1	0.1	0.1	12	0.1
Bathophenanthroline1662-01-7	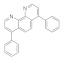	0.09	0.019	0.03	0.01	1.5	25	3	13
2,9-Dimethyl-4,7-diphenyl-1,10-phenanthroline4733-39-5	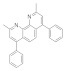	25	0.78	50	50	0.1	0.1	>100	>100
Bathocuproinedisulfonic acid sodium salt1257642-74-2	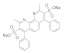	>100	>100	0.19	1.2	50	0.1	>100	>100

**Table 8 antibiotics-10-00337-t008:** Summary of anti-bacterial activity and toxicity of compounds inspired by the natural products characterised in this project.

Sample Name, structure and CAS no	Minimum Inhibitory/Toxicity Concentration of Drug (µM)
MRSA	*E. coli*	*P. aeruginosa*	*M. tuberculosis H37Rv*	HepG2	HEK 293	A549	THP-1
Hexylamine 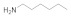 111-26-2	>100	>100	50	>100	>100	>100	>100	>100
Dihexylamine 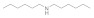 143-16-8	12.5	12.5	6.2	>100	>100	100	>100	>100
Dioctylamine 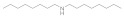 1120-48-5	6.2	6.2	3	>100	>100	>100	>100	>100
Triethylamine 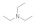 121-44-8	>100	>100	>100	>100	>100	>100	>100	>100
Tributylamine 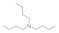 102-82-9	>100	>100	50	>100	>100	>100	>100	>100
Trihexylamine 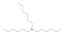 102-86-3	3	3	6.5	>100	>100	50	100	>100
Trioctylamine 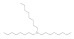 1116-76-3	12.5	25	25	0.02	>100	6	50	3.1
Benzylamine 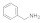 100-46-9	>100	>100	>100	>100	>100	>100	>100	>100
Dicyclohexylamine 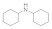 101-83-7	>100	>100	12.5	>100	>100	>100	>100	>100
Dibenzylamine 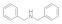 103-49-1	>100	>100	12.5	>100	>100	>100	>100	>100
Tribenzylamine 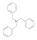 620-40-6	>100	>100	6.5	>100	>100	>100	>100	>100
2,2′-Bipyridyl 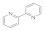 366-18-7	>100	>100	>100	0.1	3	3	>100	50
Isoquinoline 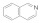 119-65-3	>100	>100	>100	0.1	3	3	>100	50
Quinoline 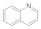 91-22-5	>100	>100	>100	0.1	3	3	50	50
4,7-phenanthroline 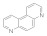 230-07-9	1.5	1.5	2.5	2.5	>100	>100	>100	>100
1,7-phenanthroline 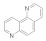 230-46-6	25	25	2.5	>100	>100	>100	>100	>100
1,10 phenanthroline 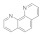 66-71-7	>100	50	0.3	0.1	0.1	0.1	50	3
4,7-DImethyl-1,10-Phenanthroline 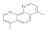 3248-05-3	0.3	0.3	0.1	0.1	0.1	0.1	0.1	0.1
Neocuproine hemihydlyate 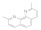 34302-69-7	0.09	0.3	0.04	0.1	0.1	0.1	6.2	12
5,6-Dimethyl-1, 10-phenanthroline 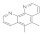 3002-81-1	0.3	0.78	>100	0.1	0.1	0.1	0.1	0.1
3,4,7,8-Tetramethyl-1,10-phenanthroline 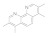 1660-93-1	0.3	0.3	0.3	0.1	0.1	0.1	0.1	0.1
1,10-Phenanthroline-5,6-dione 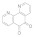 27318-90-7	0.04	0.04	0.04	0.1	0.1	0.1	0.1	0.1
5-Nitro-1,10-phenanthroline 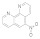 4199-88-6	0.09	0.78	>100	0.1	0.1	0.1	12	0.1
Bathophenanthroline 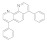 1662-01-7	0.09	0.019	0.03	0.01	1.5	25	3	13
2,9-Dimethyl-1,7-diphenyl-1,10-phenatroline 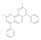 4733-39-5	25	0.78	50	50	0.1	0.1	>100	>100
Bathocuproinedisulfonic acid sodium salt 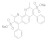 1257642-74-2	>100	>100	0.19	1.2	50	0.1	>100	>100

## Data Availability

Not applicable.

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
