# Peer review of "Identification of Bioactive Compounds from Marine Natural Products and Exploration of Structure-Activity Relationships (SAR)"

_antibiotics, 2021, doi:10.3390/antibiotics10030337_

Round 1

Reviewer 1 Report

The topic of the manuscript is interesting. It can be accepted after some amendments.

(1) In the cytotoxic study, why no non-transformed cells were used? The authors should identify if these compounds have selectivity or not.

(2) The chemical structures of the compounds with strong potencies should be displayed. 

(3) Any known compound with similar structure and biological activities? If yes, their structure should be shown. Also, they should be included into the cyto-toxic study. 

Author Response

Dear reviewer,

I sincerely appreciate your taking the time to review the manuscript.

I am grateful to hear your viewpoints and I have considered your insightful comments to improve my manuscript. Those changes are highlighted within the manuscript.

Thanks

Mojdeh

Reviewer corrections recommended.

Corrections recommended

Response

In the cytotoxic study, why no non-transformed cells were used? The authors should identify if these compounds have selectivity or not.

In this study we used

-Human alveolar epithelial cells (A549)

-Madin-Darby canine kidney epithelial cells (MDCK)

-Human leukaemia cells (THP-1)

-Human hepatocellular carcinoma cells (Hep-G2)

-Human embryonic kidney cells 293 (HEK293)

 All these cell lines grow as a monolayer when attached to culture flask. They collected from carcinoma and normal cell. These cell lines chosen because they are widely used in different fields of medical research, especially basic cancer research and drug discovery. Due to low amounts of compound isolated, we couldn’t run selectivity test. But selectivity, cyto-toxic against the cell lines and primary cells, as well as pharmacodynamic functions of proposed structures confirmed through analysis of the synthetic derivatives. This part is published doi: 10.3389/fmicb.2020.551189

The chemical structures of the compounds with strong potencies should be displayed.

We elucidated chemical structures of strong biological activities compound 1-8. They displayed in Figure 2. Line 178-180.

Any known compound with similar structure and biological activities? If yes, their structure should be shown. Also, they should be included into the cyto-toxic study.

The structure–activity relationships (SAR) has shown the modification of existing molecules can be used to produce new antimicrobial molecules and improve the biological activity by increasing antibacterial activity and decreasing toxicity. In the bellow tables antimicrobial activity and cyto-toxicity of similar structure presented.

Table 3. Screening of commercially available amines with structures similar to 3 and 13. Lines 250.

Table 4. Screening of commercially available amines with structures related to 1a, 1b and 2a. Lines 256.

Table 5. Screening of commercially available compounds with structures inspired by ascididemin. Line 315.

Table 6. Screening of compounds with structures inspired by ascididemin which include three aromatic rings. Line 353.

Table 7. Screening of compounds with structures inspired by ascididemin which include three aromatic rings plus further modifications. Line 420.

Table 8 present summary of anti-bacterial activity and toxicity of compounds inspired by the natural products characterised in this project. Line 439.

Reviewer 2 Report

I am glad to review this research work done by Dinarvand et. al. This is a good piece of work; however, the author needs to work on the presentation of the manuscript, some part of the manuscript could move to supporting information, and the activity profile could move from supporting information to the manuscript. I would like to recommend the article could be published in Antibiotics, with minor revision.

The authors could make the following minor changes.

  1. The author could provide a structure of methicillin, vancomycin, ascididemin, quinoline, phenanthroline, and ω-conotoxin MVIIA
  2. The author could provide the structure of the most potent reported marine natural products having antibacterial activities.
  3. The author could provide the structures of 7, 8, and 9 along with 1-5 in figure 1. If possible, for all the compounds discussed in this manuscript (eq. isoquinoline, quinoline, 4,7- 158 phenanthroline, 1,7-phenanthroline, 1,10-phenanthroline, and bipyridine).
  4. The author could change “(j + k) = 14; m =5, n = 9; (q +r) = 20; (t + u + v) = 19” to simplified equation; like j = 7…..
  5. Any reason why cyclic amines are not active compare to cyclic amine?
  6. Any activity profile for Benzyl amine having electron-withdrawing/or donating groups.
  1. The author could move “Purification of Natural Products from Extracts and Structure Elucidation” and “Identification and Structure Elucidation” to the supporting information and move “STRUCTURE−ACTIVITY RELATIONSHIP” from supporting information to the manuscript.
  2. The author could show structures while discussing the activity of series of compounds and then show the activity profile (MIC) in the table.
  3. Any comparison of the activity of different series of compounds.
  4. In line 315: “1H NMR (500 MHz, DMSO-d6)” it should not be bold.
  5. .In Line 320: Use the abbreviation of “rdbe”.
  1. In Line 247: Change “37°C” to “37 °C”.
  2. For the references use the same format: need space and year should be in bold. Change “2014,20(4), 3-18” to 2014, 20(4), 3- 18” and make changes for the other references as well. In Line 517: change “Mar Drugs 2014,12 (1), 462-76.” To “Mar Drugs 2014,12 (1), 462-476”. And make changes for the other references as well.
  3. In Table S5, the author should draw the COSY relationship in structure 6 with arrows for better understanding as like compound 8 (Fig S25).
  4. All NMR should have pick peaking, integration with the structure embedded.
  5. Table S8-12 should move to the manuscript instead of supporting documents.

Author Response

Dear reviewer,

I sincerely appreciate your taking the time to review the manuscript.

I am grateful to hear your viewpoints and I have considered your insightful comments to improve my manuscript. Those changes are highlighted within the manuscript.

Thanks

Mojdeh

Reviewer corrections recommended.

Corrections recommended

Response

The author could provide a structure of methicillin, vancomycin, ascididemin, quinoline, phenanthroline, and ω-conotoxin MVIIA

In this work we present the structures that we discovered and re discovered from marine natural products as well as, commercially available compounds with structures similar to synthesised compound which inspired by natural product structures. Structures of ascididemin (line 170-175), quinoline (lines 284,320) and phenanthroline (lines 284,324-335) are provided in this work.  The structures of methicillin, vancomycin and ω-conotoxin MVIIA did not present because structurally, they are not close to natural product structures or synthesised compound.  Also, the activities and cyto-toxicity of methicillin and ω-conotoxin MVIIA didn’t test.  We cannot compare they activities with isolated natural product. The vancomycin only used as a positive control.

The author could provide the structure of the most potent reported marine natural products having antibacterial activities.

Change done. Lines 69-81

The author could provide the structures of 7, 8, and 9 along with 1-5 in figure 1. If possible, for all the compounds discussed in this manuscript (eq. isoquinoline, quinoline, 4,7- 158 phenanthroline, 1,7-phenanthroline, 1,10-phenanthroline, and bipyridine).

Change done.

lines 167-176

lines 215 -222

lines 234-237

lines 281-285

lines 299-309

lines 382-385

The author could change “(j + k) = 14; m =5, n = 9; (q +r) = 20; (t + 5u + v) = 19” to simplified equation; like j = 7…..

Change done.

lines 178

Any reason why cyclic amines are not active compare to cyclic amine?

All the cyclic amines reduced the activities, and they are not toxic. We only observed two different result for two Gram negative bacteria. This phenomenon can explain via their patterns of susceptibility and growth rate.

Any activity profile for Benzyl amine having electron-withdrawing/or donating groups.

We didn’t consider other positions of disubstituted benzene ring to test the effect of an electron donating groups on molecules antimicrobial activities.  The main aim of this study was to modify structure of molecules by adding or removing parts/atoms (C, O, N..).  The chemically modified electrodes were partially covered.

The author could move “Purification of Natural Products from Extracts and Structure Elucidation” and “Identification and Structure Elucidation” to the supporting information and move “STRUCTURE−ACTIVITY RELATIONSHIP” from supporting information to the manuscript.

Change done.

Structure−Activity Relationship results moved to the manuscript.

Table 3. Screening of commercially available amines with structures similar to 3 and 13. Lines 250.

Table 4. Screening of commercially available amines with structures related to 1a, 1b and 2a. Lines 256.

Table 5. Screening of commercially available compounds with structures inspired by ascididemin. Line 315.

Table 6. Screening of compounds with structures inspired by ascididemin which include three aromatic rings. Line 353.

Table 7. Screening of compounds with structures inspired by ascididemin which include three aromatic rings plus further modifications. Line 420.

Table 8 present summary of anti-bacterial activity and toxicity of compounds inspired by the natural products characterised in this project. Line 439.

Purification of Natural Products from Extracts and Structure Elucidation” and “Identification and Structure Elucidation” to the supporting information are the main parts of methodology, if this is possible, we keep them in the manuscript, please.

The author could show structures while discussing the activity of series of compounds and then show the activity profile (MIC) in the table.

Change done.  

Table 8 present summary of anti-bacterial activity and toxicity of compounds inspired by the natural products characterised in this project. Compounds structures added to Table. Line 439.

Any comparison of the activity of different series of compounds.

In this study we work with Gram positive, Gram-negative and mycobacteria pathogens. Modification of molecule shown different effect on test pathogens. In summary, we reported by varying the number of carbons in the linear chains, the bioactivity against the Gram positive and Gram-negative pathogens tested increased. By extending the carbon linker to the aromatic rings then modifying and increase the aromatic rings activity of the molecules against tested pathogens reduced. This suggests that the mechanism of action may be based on targeting complex cell wall structures. Many factors influence the activity of a molecule, including side chain length, symmetry, length of the alkyl chain, the presence of charged pyridinium moiety, presence and position of the carboxylate in the structure, position of the methyl group, distance between the two pyridinium moieties and chain linker between the two pyridinium rings. Thus, the mechanism of action of these molecules may be based on the space between the rings and distance between charged moieties, which possibly manifests through interaction with membrane sites. In the next step of my work, I synthesised antimicrobial compound with fluorescent activity similar to natural product structure to track the molecule in the live cell and find the  compound  mechanism of action. I will publish that part separately.

In line 315: “1H NMR (500 MHz, DMSO-d6)” it should not be bold.

Change done.

line 530

In Line 320: Use the abbreviation of “rdbe”.

rdbe is abbreviation of ring or double bond equivalent.

This definition presented in line 137.

In Line 247: Change “37°C” to “37 °C”.

Change done. line 462

For the references use the same format: need space and year should be in bold. Change “2014,20(4), 3-18” to 2014, 20(4), 3- 18” and make changes for the other references as well. In Line 517: change “Mar Drugs 2014,12 (1), 462-76.” To “Mar Drugs 2014,12 (1), 462-476”. And make changes for the other references as well.

Change done.

Lines 624-780

In Table S5, the author should draw the COSY relationship in structure 6 with arrows for better understanding as like compound 8 (Fig S25).

The arrows in the Fig S25 are based on the HMBC NMR spectrum analysis.  Due to low amounts of compound isolated we didn’t run HMBC for compound 6 and 7.

All NMR should have pick peaking, integration with the structure embedded.

Change done.  Fig S27. Also, since, mass spectrometry (MS) used in this study provides an exhaustive strategy to elucidate chemical structures from insufficient resources such as natural products. I put compound 6, 7 and 8 mass patterns with the structure embedded. Fig S26

Table S8-12 should move to the manuscript instead of supporting documents.

Change done.

Table 3. Screening of commercially available amines with structures similar to 3 and 13. Lines 250.

Table 4. Screening of commercially available amines with structures related to 1a, 1b and 2a. Lines 256.

Table 5. Screening of commercially available compounds with structures inspired by ascididemin. Line 315.

Table 6. Screening of compounds with structures inspired by ascididemin which include three aromatic rings. Line 353.

Table 7. Screening of compounds with structures inspired by ascididemin which include three aromatic rings plus further modifications. Line 420.

This manuscript is a resubmission of an earlier submission. The following is a list of the peer review reports and author responses from that submission.

Round 1

Reviewer 1 Report

The manuscript is apparently aimed to investigate marine sponge extracts for new bioactive compounds. There is however a misleading title citing “marine microorganism”! I also note that in the line 19 of the abstract is reported “MICs from 15.6 to 62.5 mg/mL”, while probably should be micro g/mL!

The experimental approach is based on the screening of a large library of marine sponge extracts (1434) obtained by the Queensland Compund Library, looking for activity against MRSA.

Although the large effort it is my opinion that results are quite poor and of very little scientific value.

A first set of compounds indicated as potentially active is made up by a group of aliphatic and aromatic amines. These molecules are not completely characterized, and biological activities are not relevant. Furthermore, it is not a novelty that amines might show antibacterial activities (Antimicrobial Agents and Chemoterapy 1972, Vol. 2, No. 6, pp 492-498; Z. Naturforsch. 1979, 34 c, pp 485-486; Bioorganic and Medicinal Chemistry 2015, vol 23, Issue 2, pp 290-296).

SAR investigation and comparison with synthetic and commercially available compounds do not add significative advancements.

The remaining described compounds 6, 7 and 8, identified as the known ascididemnin, 2-bromoascididemnin and halisulfate respectively, might be slightly more interesting. However, these molecules are largely known and investigated. Furthermore, in the manuscript are missing 13C reference data for complete structure identification and evaluation of level of purity of sample submitted to biological assays.

To my opinion, the manuscript is largely unfit for publication.

Author Response

Dear reviewers

I sincerely appreciate your taking the time to review the manuscript.

I am grateful to hear your viewpoint and I have considered your insightful comments to improve my manuscript. Those changes are highlighted within the manuscript.

Thanks

Mojdeh

Reviewer 2 Report

Please see the attchment.

Author Response

(The authors gave the same response as above.)

Reviewer 3 Report

My considerations are in the attached file.

In your next submissions, be sure to submit a file with a minimum of quality in the formatting of the text.

Author Response

(The authors gave the same response as above.)

Round 2

Reviewer 1 Report

In supplementary material, pages 19-24, there are several tables with indication of "1H (500 MHz) and 13C (125 MHz) NMR spectroscopic data". However 13C (125 MHz) NMR data are always missing. These should be provided.

Manuscript revision is appropriate.

Reviewer 2 Report

The authors have responded to my comments adequately. I agree to accept this revised manuscript in current version.

Reviewer 3 Report

In all my research experience, I have never seen - and I do not believe - that a study of this complexity, involving refined and elaborate spectrometric procedures such as tandem-MS and NMR, biological tests and SAR studies have been carried out by only 1 author (as described in the contributions by author at the end of the manuscript). 
It is impossible to dominate these areas individually.

In my evaluation, the manuscript is not well written.

Although the authors insist on talking that the results indicate potential activity, the experimental data - besides being very basic, requiring many other tests - are not at all encouraging. 

In addition, the prediction of activity is a very "forced" extrapolation.

Thus, I maintain my position of declining the manuscript for publication.